# Advances in Pretreatment Methods for Free Nucleic Acid Removal in Wastewater Samples: Enhancing Accuracy in Pathogenic Detection and Future Directions

Kien A. Vu [1],*[ID], Thu A. Nguyen [2][ID] and Thao P. Nguyen [3]

[1] Department of Civil and Environmental Engineering and Earth Sciences, University of Notre Dame, Notre Dame, IN 46556, USA
[2] Sol Price School of Public Policy, University of Southern California, Los Angeles, CA 90089, USA; thuanguy@usc.edu
[3] Department of Computer Science, University of Calgary, Calgary, AB T2N1N4, Canada
* Correspondence: avu2@nd.edu

**Abstract:** Accurate pathogenic detection in wastewater is critical for safeguarding public health and the environment. However, the presence of free nucleic acids in wastewater samples poses significant challenges to molecular detection accuracy. This comprehensive review explores the current status and future potential of pretreatment methods to remove free nucleic acids from wastewater samples. The study contributes a comprehensive analysis of the mechanisms, strengths, and limitations of various pretreatment approaches, including physical, chemical, and enzymatic processes. The effect of various factors on the removal efficiency of these pretreatment methods is also discussed. This review enhances our comprehension of pretreatment techniques and their vital role in achieving precise pathogenic detection in complex wastewater matrices. Furthermore, it outlines future perspectives and developments for improving the speed and effectiveness of pathogenic detection, contributing significantly to disease surveillance, early warning systems, and environmental protection.

**Keywords:** pathogenic detection; wastewater; free nucleic acids; pretreatment methods; environmental analysis; nucleic acid removal





## 1. Introduction

Wastewater often contains a diverse range of pathogens, including bacteria, viruses, and fungi, due to human and animal waste, industrial discharges, and runoff from contaminated areas [1]. The existence and abundance of pathogens in wastewater raise significant concerns for public health and the environment. Pathogens in wastewater can cause waterborne diseases, which will produce severe illnesses such as cholera, typhoid, hepatitis, and gastroenteritis [2]. Moreover, the release of untreated or partially treated wastewater into natural water bodies can contaminate aquatic ecosystems and disrupt the balance of natural environments [3,4].

Pathogenic detection in wastewater samples plays a key role in safeguarding public health and the environment. Accurate pathogenic detection is not only essential for preventing the spread of waterborne diseases, but also for preserving the ecological integrity of natural water bodies [5,6]. It is also vital to monitor and detect pathogens in wastewater samples for assessing the effectiveness of wastewater treatment processes, which will help in determining the sufficiency of treatment facilities in eliminating pathogens before discharging treated wastewater into the environment. Pathogenic detection in wastewater can serve as an early warning system for potential disease outbreaks [7]. By monitoring the presence of specific pathogens in wastewater, proactive measures can be taken to protect public health and mitigate the spread of diseases [8].

Traditional techniques, such as culture-based methods, microscopic observation, and filtration, have been used to identify and quantify pathogens in wastewater, forming

the foundation of early strategies for waterborne pathogen detection [9]. Culture-based approaches involve the cultivation of microorganisms on specific media, enabling their identification and characterization and contributing to a comprehensive understanding of water quality. The microscopic method focuses on the direct visualization of pathogens under a microscope, offering insights into their morphology and abundance. Filtration techniques concentrate pathogens from high water amounts through filters to capture microorganisms, followed by elution and further analysis.

Meanwhile, molecular methods have been widely utilized for pathogen detection in wastewater. In contrast to culture-based approaches, molecular techniques exhibit higher sensitivity and the capacity to detect low concentrations of target pathogens in wastewater [10]. Their significantly reduced detection time, often within a few hours, makes them suitable for time-sensitive applications [10]. Moreover, these methods enable the precise quantification of target DNA, providing valuable insights into pathogen abundance [11].

The precision and reliability of pathogenic detection in wastewater are challenged by the presence of free nucleic acids, primarily DNA and RNA. These free nucleic acids, originating from both viable and non-viable microorganisms residing in wastewater, introduce complexities that compromise the accuracy of detection and quantification [12]. To address these challenges, various pretreatment methods have been developed and employed to selectively remove or reduce the levels of free nucleic acids in wastewater samples [13]. These pretreatment methods can mitigate interference from nontarget nucleic acids, thereby enhancing the sensitivity of pathogen detection techniques and enabling the identification of low pathogen concentrations.

The objective of this review is to provide an analysis of various pretreatment approaches, such as physical, chemical, and enzymatic methods, that have been applied to wastewater samples. Through a critical evaluation of the strengths and limitations of these methods, we aim to clarify their role in achieving accurate pathogenic detection and their potential for addressing the challenges posed by free nucleic acids. This review explores the applications of pretreatment methods across diverse wastewater types, the effect of some factors on pretreatment methods, and the future prospects of wastewater pathogenic detection with improved pretreatment methodologies.

## 2. Free Nucleic Acids in Wastewater Samples

Wastewater refers to water that has been used in various human and industrial activities, including domestic, industrial, and agricultural processes (Table 1). It contains a mixture of contaminants, such as organic and inorganic substances, suspended solids, and microorganisms. Free nucleic acids in wastewater originate from diverse sources, for example, human and animal manure, microbial processes, plant materials, and industrial activities [14,15]. Consequently, they exhibit remarkable diversity in terms of genetic material, species, and functionality. Due to their stability, free nucleic acids can persist in the environment for a long time, influencing genetic material analysis and extending the potential for pathogenic threats [16].

**Table 1.** Characteristics and common pathogens of some wastewaters.

| Type | Component | Source | Main Pathogen | Reference |
|---|---|---|---|---|
| Municipal wastewater | - Human waste, <br> - Greywater, blackwater <br> - Detergents, food scraps, paper <br> - Other household wastes | - Residences <br> - Businesses <br> - Schools <br> - Hospitals <br> - Institutions | - Bacteria (*E. coli, Salmonella, Campylobacter*) <br> - Enteric viruses <br> - Protozoa (*Giardia, Cryptosporidium*) | [17] |

**Table 1.** *Cont.*

| Type | Component | Source | Main Pathogen | Reference |
|---|---|---|---|---|
| Industrial wastewater | - Chemicals<br>- Heavy metals<br>- Organic compounds<br>- Oils | - Factories<br>- Refineries<br>- Processing plants<br>- Manufacturing<br>- Facilities industrial operations | - Bacteria (*E. coli, Salmonella, Shigella*)<br>- Viruses (norovirus, rotavirus)<br>- Protozoa (*Giardia, Cryptosporidium*) | [6] |
| Agricultural wastewater | - Pesticides<br>- Herbicides<br>- Fertilizers<br>- Animal waste<br>- Soil particles<br>- Organic matter | - Farms<br>- Crop fields<br>- Livestock operations<br>- Agricultural activities | - Bacteria (fecal coliforms, *E. coli*)<br>- Parasites (helminths)<br>- Zoonotic pathogens. | [18] |
| Hospital wastewater | - Infectious agents (bacteria, viruses, fungi)<br>- Pharmaceutical residues<br>- Biohazardous waste<br>- Disinfectants<br>- Chemicals | - Hospitals<br>- Clinics<br>- Healthcare facilities<br>- Laboratories | - Antibiotic resistant bacteria (MRSA)<br>- Bloodborne pathogens (Hepatitis B and C)<br>- Disease-causing viruses. | [19] |
| Stormwater runoff | - Chemicals (oil, heavy metals, fertilizers)<br>- Sediments<br>- Debris<br>- Contaminants from urban environment | - Urban and suburban areas where rainfall or snowmelt flows over impervious surfaces | - Bacteria (*E. coli, Enterococci*)<br>- Viruses (norovirus, rotavirus)<br>- Protozoa (*Giardia, Cryptosporidium*) | [20] |
| Leachate | - Heavy metals<br>- Organic compounds<br>- Chemical residues | - Landfills and sites where waste materials decompose and interact with moisture | - Bacteria (*E. coli*)<br>- Viruses (enteroviruses, adenoviruses)<br>- Parasites (*Giardia, Cryptosporidium*) | [21] |

Free nucleic acids can interfere with the common molecular detection methods used for pathogen identification in wastewater, including polymerase chain reaction (PCR), reverse transcription-PCR (RT-PCR), and next-generation sequencing (NGS). Specifically, the presence of extraneous genetic material can compete with or inhibit the amplification of the target pathogen's DNA or RNA, potentially resulting in false positive or negative outcomes [22]. A high concentration of free nucleic acids can dilute the target pathogen's genetic material within a sample, reducing the sensitivity of detection methods and making it more difficult to identify low pathogen levels in wastewater. Moreover, free nucleic acids can act as potential sources of cross-contamination in laboratories, leading to inaccurate results [23].

There is also the risk of misidentification, where free nucleic acids are occasionally detected as pathogenic genetic material in molecular assays, potentially resulting in false positives [24]. Furthermore, the broader genetic diversity present in wastewater can make it challenging to distinguish between closely related microorganisms or identify specific strains of pathogens. Thus, the removal or reduction of free nucleic acids is essential to enhance target pathogen recovery and improve the reliability of detection techniques.

## 3. Pretreatment Methods

### 3.1. Overview of Pretreatment Techniques

Pretreatment methods are designed to selectively remove or reduce the levels of free nucleic acids in wastewater samples. By minimizing the existence of interfering genetic material, pretreatment methods amplify the sensitivity of pathogenic detection, enabling the identification of pathogens even at low concentrations within complex wastewater matrices. These techniques promote the specificity of pathogenic detection assays by diminishing background genetic material, thus facilitating the differentiation between closely related

microorganisms or specific pathogenic strains [25]. Effective pretreatment minimizes the risk of cross-contamination in the laboratory to guarantee the precision and reliability of the results. Additionally, pretreatment techniques contribute to the standardization of the pathogenic detection process, making it more consistent across diverse wastewater samples characterized by varying levels of interfering nucleic acids [26,27].

In molecular detection methods, such as PCR, pretreatment is typically conducted during the sample preparation stage (Figure 1). At this stage, collected wastewater samples may contain a complex mixture of substances, including free nucleic acids, contaminants, and pathogens. Solid particles and debris are separated from the liquid fraction of the sample through filtration. Subsequently, chemical or enzymatic pretreatment methods may be employed to disrupt the structure of cells and pathogens within the sample, thereby releasing genetic material (DNA or RNA) into the solution. Following cell lysis, the sample solution contains a mixture of nucleic acids, including both the genetic material of the target pathogen and interfering free nucleic acids. To selectively isolate the nucleic acids from other components, various extraction methods, such as spin column-based extraction, magnetic bead-based extraction, or organic extraction, have been applied [28–30].

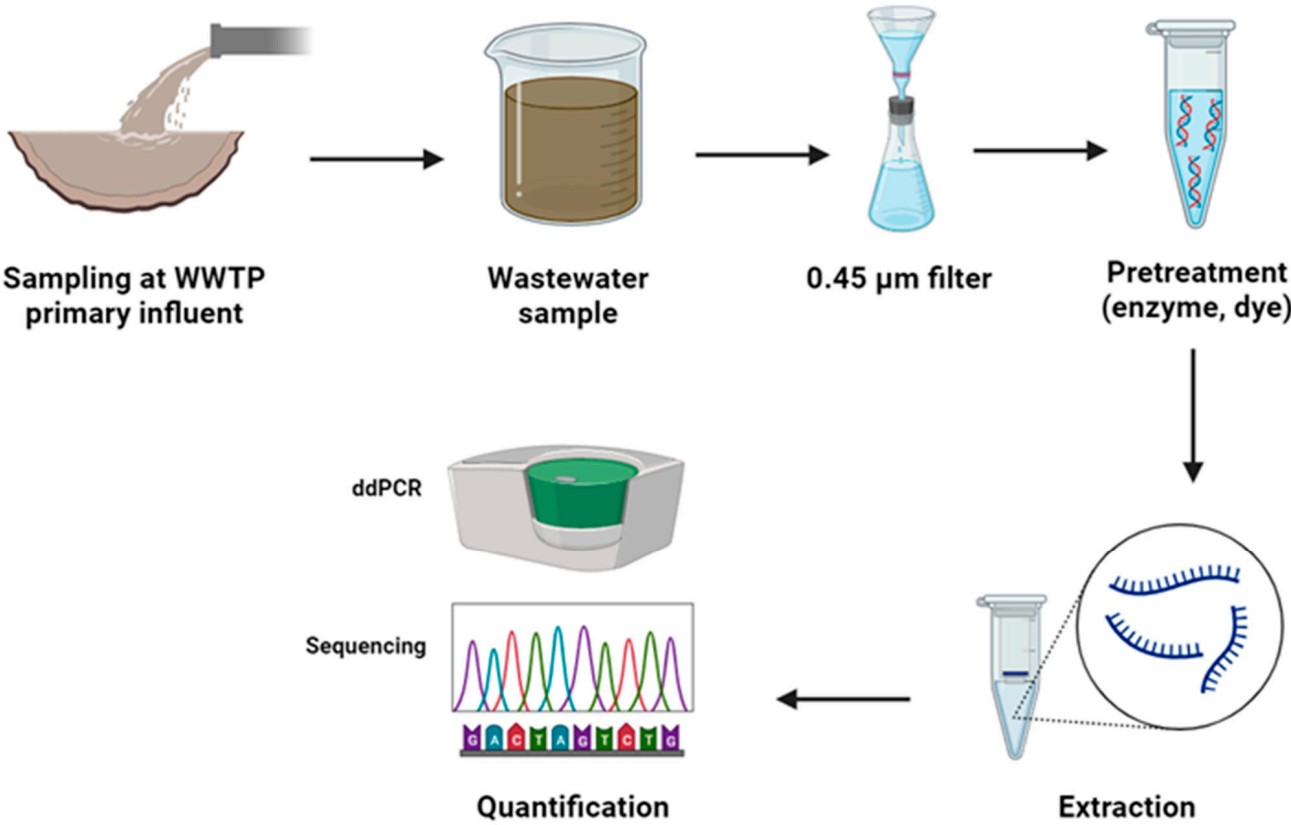

**Figure 1.** Pretreatment methods in the molecular detection methods using PCR techniques.

After pretreatment and extraction, the concentration of the target pathogen's genetic material can be detected using diverse methods, such as sequencing and real-time PCR. The obtained results are evaluated to detect the presence or absence of the target pathogen. The data derived from the PCR analysis are interpreted, and the quantification of the target pathogen is determined based on the assay's specific parameters and the standards established during the pretreatment and PCR setup stages.

*3.2. Physical Pretreatment Methods*

3.2.1. Filtration

Filtration, such as microfiltration and ultrafiltration, is a common pretreatment method to remove solid particles, suspended solids, and other particulate matter from wastewater

samples. This step plays a crucial role in sample preparation by ensuring that undesired particles or contaminants do not interfere with subsequent molecular analyses like PCR, qPCR, and sequencing [27]. Moreover, filtration can be effectively utilized to concentrate microorganisms and pathogens, simplifying their detection and analysis within the sample. Notably, filtration efficiently removes contaminants, including organic and inorganic particles, that may hinder molecular detection assays [31].

Filtration pretreatment encompasses sample preparation, filter setup, sample loading, filtration, recovery, and analyte analysis [32]. In this context, wastewater samples are initially collected and prepared for filtration. A filtration apparatus, which includes a filter membrane, a filter holder, and a vacuum or pressure source, is assembled. The wastewater sample is then loaded onto the filter membrane, designed to retain particles while allowing the filtrate to pass through. A vacuum or pressure source is employed to force the sample through the filter membrane. Consequently, retained particles, debris, or microorganisms are collected on the filter, while the clarified filtrate is separately obtained. The retained particles or concentrated analytes may be recovered from the filter membrane through techniques like scraping, elution, and dissolution. The clarified filtrate or recovered analytes are ready for molecular detection methods.

### 3.2.2. Centrifugation

Centrifugation involves using centrifugal force to separate particles, cells, or analytes from the liquid phase within a sample (Figure 2). It serves the purpose of concentrating cells, microorganisms, and pathogens, simplifying their detection and subsequent analysis in molecular assays. Centrifugation is a key step in many nucleic acid isolation protocols, facilitating the separation of DNA or RNA from other components within the sample [33].

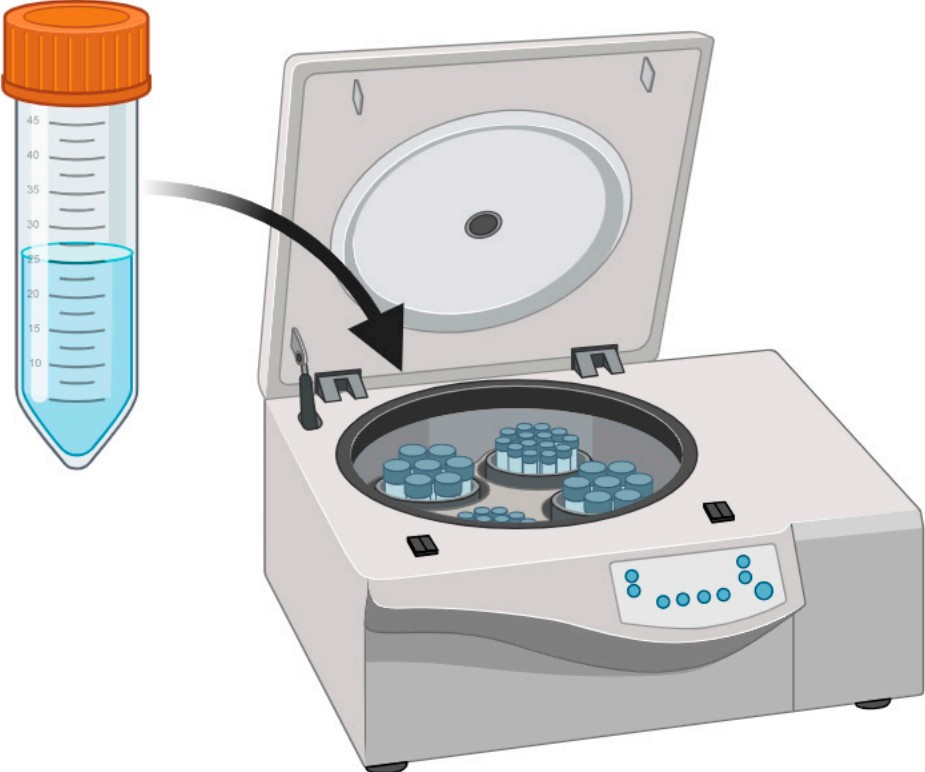

**Figure 2.** Centrifugation pretreatment technique.

There are two main types of centrifugation for pretreatment in pathogenic detection. Differential centrifugation involves a sequence of centrifugation steps at varying speeds to separate particles and components of different sizes and densities. This method is frequently utilized to isolate cells, microorganisms, and other analytes from complex

samples. In contrast, gradient centrifugation adopts density gradients, typically sucrose or cesium chloride, to separate analytes based on density. It offers a more precise separation and purification of analytes [34].

The general procedure for centrifugation includes sample preparation, sample loading, centrifugation, supernatant collection, pellet resuspension, and analyte analysis [35]. Wastewater samples are collected and loaded into centrifuge tubes or containers. The tubes are placed in a centrifuge, where particles and analytes are separated by centrifugal force based on density and size. After centrifugation, the supernatant (liquid phase) is carefully removed, leaving behind a pellet that contains concentrated analytes or particles. The pellet may be resuspended in an appropriate buffer or solvent for further analysis. The concentrated analytes or pelleted materials are ready for molecular detection methods.

### 3.2.3. Ultrasonication

Ultrasonication is a valuable pretreatment method for pathogenic detection in molecular analyses like PCR, qPCR, or sequencing. It uses high-frequency sound waves (ultrasound) to disrupt cells, tissues, or particles, thereby facilitating the release of nucleic acids from complex matrices. The mechanism of ultrasonication relies on the formation and rapid collapse of tiny bubbles within a liquid exposed to high-frequency sound waves [33]. This phenomenon generates localized high-energy forces capable of disrupting cellular membranes, breaking down tissues, and releasing analytes from the sample matrix.

Ultrasonication involves sample preparation, sample loading, ultrasonication, sample cleanup, and analyte recovery. Specifically, wastewater samples are collected and prepared through filtration, dilution, or concentration. The prepared sample is placed in a microcentrifuge or extraction tube, which will be placed in an ultrasonication bath or a probe-type sonicator. High-frequency sound waves are applied to the sample to disrupt cells, tissues, or particles and release analytes. After ultrasonication, the released analytes may undergo additional purification or cleanup steps. The disrupted sample is ready for molecular detection techniques [36].

### 3.3. Chemical Pretreatment Methods

### 3.3.1. Precipitation

Precipitation methods can be employed to concentrate and purify DNA or RNA from wastewater samples. They can effectively remove contaminants, impurities, and interfering substances that may affect the quality and accuracy of molecular detection [33]. It encompasses sample preparation, precipitation reagent addition, mixing and incubation, centrifugation or filtration, supernatant removal, and resuspension [37]. In particular, wastewater samples are collected, filtered, diluted, or concentrated. An appropriate precipitation reagent, such as alcohol or salt, is added to the wastewater sample to initiate the precipitation process. The sample is mixed to ensure the proper distribution of the precipitation reagent and analyte, and then incubated at specific conditions to allow the analyte to aggregate and precipitate. Centrifugation or filtration will be subsequently employed to separate the precipitate from the liquid phase. The liquid phase (supernatant) is carefully removed, leaving behind the precipitate, which may be resuspended in an appropriate buffer or solvent for further analysis.

### 3.3.2. Organic Extraction

Organic extraction can be applied to isolate and purify nucleic acids from wastewater samples. This method involves using organic solvents to selectively separate the nucleic acids from the sample matrix, thereby improving the quality and accuracy of molecular detection [38]. It comprises sample collection and preparation, organic solvent addition, mixing and phase separation, collection of organic phase, concentration, and resuspension. After the filtration, dilution, or concentration process, an appropriate organic solvent, such as ethanol or isopropanol, is added to the wastewater sample to initiate the extraction process. The sample is mixed to ensure proper contact between the solvent and the analyte.

Phase separation, often facilitated by centrifugation or gravity, is then performed to separate the desired analyte from the aqueous matrix. The phase containing the target analyte is carefully collected, while the undesired aqueous phase containing impurities and contaminants is discarded. If the analyte is in low concentrations, the collected phase may undergo concentration steps to increase the analyte's yield. The isolated and purified analyte is resuspended in an appropriate buffer or solvent for subsequent molecular detection [39].

### 3.3.3. Dye Treatment Method

Dye-based pretreatment methods, such as propidium monoazide (PMA), are used to selectively inhibit the detection of non-viable cells or DNA from dead microorganisms in wastewater samples (Figure 3).

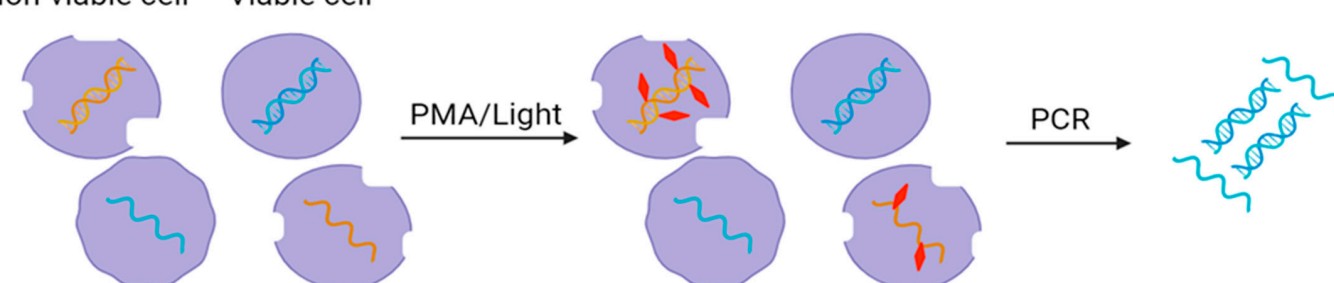

**Figure 3.** Use of PMA dye (red) for removing free nucleic acids in wastewater.

Dyes have the ability to permeate the membranes of dead or non-viable cells, but remain unable to enter the intact membranes of live or viable cells. Once inside the non-viable cell, the dye intercalates with the DNA, forming a covalent bond upon exposure to a light source. This covalent bonding effectively impedes DNA amplification during molecular detection methods like PCR, qPCR, or sequencing [40]. The outcome is the selective inhibition of DNA amplification from non-viable microorganisms, ensuring that only DNA from vital and intact cells is detected in subsequent analyses. By excluding DNA from non-viable microorganisms, dye-based pretreatment enhances the precision of pathogenic detection in wastewater samples.

In dye-based pretreatment, wastewater samples are collected and prepared for analysis through filtration or concentration. Dye is added to the wastewater sample, permeating the cell membranes of all microorganisms, both non-viable and viable. The sample is incubated in the dark, allowing the dye to penetrate cells and specifically bind to DNA from non-viable cells. After incubation, the sample is exposed to a light source, such as a blue LED, triggering the generation of covalent bonds between the dye and DNA within non-viable cells. DNA extraction is performed on the pretreated sample, and the dye-bound DNA from non-viable cells is purposefully excluded during this extraction. The DNA extracted from viable cells is then subjected to molecular detection methods, offering a highly accurate means of pathogenic detection [41].

### 3.4. Enzymatic Pretreatment Methods

Enzymatic pretreatment methods involve the application of specific enzymes to modify or degrade particular components within the wastewater samples. The selection of the enzyme depends on the analytical objectives, such as the removal of extracellular nucleic acids (e.g., DNase or RNase) (Figure 4), the degradation of lipids or proteins (e.g., lipases or proteases), or the breakdown of complex structures (e.g., cellulases) [42].

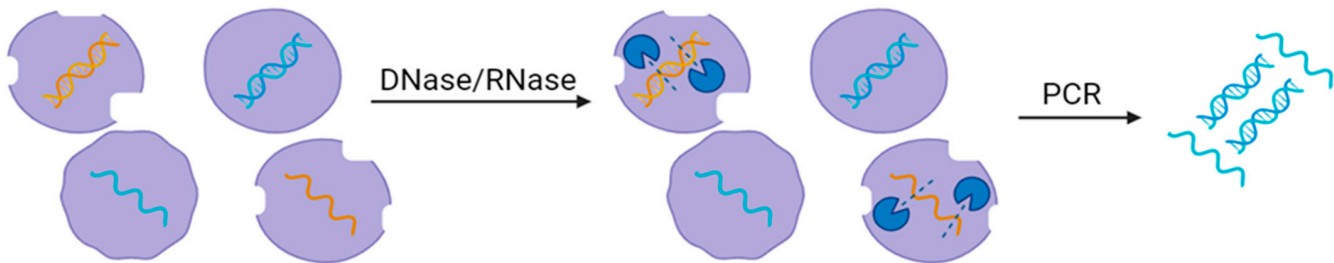

**Figure 4.** Use of DNase and RNase enzyme (blue) to remove free nucleic acids in wastewater.

Enzymatic pretreatment includes sample collection, enzyme addition, incubation, enzyme inactivation, sample cleanup, analyte extraction, and molecular detection [35]. After the filtration process, a specific enzyme is added and mixed with the wastewater sample to ensure uniform distribution. The sample is incubated, permitting the enzyme to act on the targeted components effectively. After incubation, the enzymatic activity is typically paused by heating or adding an enzyme-inactivating reagent. Additional steps for sample cleanup may be performed to eliminate any residual enzymatic activity or reaction byproducts. The treated sample is then subjected to analyte extraction and molecular detection.

## 4. Effect of Some Factors on Pretreatment Methods

### 4.1. Wastewater Type and Composition

The selection of an appropriate pretreatment method is influenced by the type and composition of the wastewater. A brief overview of the impact of common wastewater types and compositions can be found in Table 2.

**Table 2.** Effect of wastewater type on pretreatment methods.

| Type | Characteristics | Pretreatment Methods | Analytical Method | Reference |
|---|---|---|---|---|
| Municipal wastewater | - Contains a mix of domestic sewage and potentially industrial effluents<br>- High organic content, suspended solids, and potential pathogens<br>- Variable composition depending on population and industrial activity | - Enzymatic methods (e.g., DNase, RNase) for nucleic acid degradation to remove interfering DNA or RNA<br>- Filtration and centrifugation for the removal of suspended solids and debris | - Quantitative Polymerase Chain Reaction (qPCR)<br>- Digital PCR (dPCR)<br>- Droplet digital PCR (ddPCR) | [42] |
| Industrial wastewater | - Varied composition based on industry type, may contain heavy metals, chemicals, and unique contaminants<br>- Highly variable in terms of pollutants and composition | - Precipitation methods to remove heavy metals and specific contaminants<br>- Enzymatic methods for industry-specific contaminants | - qPCR<br>- dPCR<br>- ddPCR | [43] |

**Table 2.** *Cont.*

| Type | Characteristics | Pretreatment Methods | Analytical Method | Reference |
|---|---|---|---|---|
| Agricultural wastewater | - Contains organic matter, nutrients, pesticides, and potentially harmful microorganisms<br>- May vary seasonally based on crop and pesticide use | - Filtration and centrifugation for removal of suspended solids<br>- Enzymatic methods to degrade extracellular DNA or RNA<br>- PMA treatment to selectively target live microorganisms | - qPCR<br>- dPCR<br>- ddPCR | [44] |
| Hospital wastewater | - Contains pharmaceuticals, chemicals, and potentially infectious agents<br>- Variable composition depending on the medical facilities connected to the wastewater system | - Filtration and centrifugation to remove particulate matter<br>- PMA treatment for enhanced pathogenic detection accuracy | - qPCR<br>- dPCR<br>- ddPCR | [45] |
| Stormwater runoff | - Carries a mix of pollutants, including debris, sediments, chemicals, and microorganisms<br>- Composition can vary widely based on weather events and urbanization | - Filtration and centrifugation are used to remove large debris and sediments<br>- Filtration and centrifugation for removal of large debris and sediments<br>- Enzymatic methods for the selective degradation of interfering substances | - qPCR<br>- dPCR<br>- ddPCR | [46] |
| Leachate | - Arises from landfills and typically contains various organic and inorganic contaminants<br>- May also contain potentially harmful microorganisms | - Filtration and enzymatic methods to remove particulate matter and degrade extracellular nucleic acids<br>- PMA treatment for selective targeting of live microorganisms | - qPCR<br>- dPCR<br>- ddPCR<br>- Next generation sequencing (NGS) | [47] |

### 4.2. Pathogen Type

The type of pathogens in wastewater samples, such as bacteria, viruses, protozoa, fungi, and parasites, directly impact pretreatment methods. These diverse pathogens exhibit distinct characteristics in cell structure, resistance to inactivation, and the presence of extracellular nucleic acids, which influence the selection of suitable pretreatment methods. Bacterial pathogens, including *E. coli* or *Salmonella*, have cell walls that may require specific pretreatment strategies to break down and release the nucleic acids for subsequent detection. Therefore, enzymatic approaches, like lysozyme treatment, filtration, and centrifugation, may be used to disrupt and remove bacterial cell walls [48,49]. Viruses are often resistant to enzymatic treatment, thereby demanding alternative techniques like PMA treatment to selectively inactivate non-viable viral genetic material while preserving viable viral DNA or RNA [50].

Furthermore, filtration and centrifugation can productively remove larger viral particles. Due to their complex life stage, protozoan and parasitic pathogens need special pretreatment, such as enzymatic methods, to break their structures and release the genetic material [51,52]. Similarly, fungal pathogens may require enzymatic pretreatment to disrupt their cell walls and release DNA. Alternatively, filtration can be employed to eliminate fungal cells and spores [53].

### 4.3. Environmental Conditions

Environmental conditions, for instance, temperature, pH, salinity, and organic matter, can influence pretreatment methods. The effect of temperature is dependent on the specific technique and sample characteristics. For instance, DNase and RNase enzymes often

exhibit optimal activity around 37 °C. If the wastewater temperature is significantly lower or higher than this optimal range, the removal efficiency of nucleic acid can be affected [54]. To ensure effective enzymatic reactions, adjustments to the enzyme incubation temperature may be necessary based on the actual sample temperature. Low temperatures, such as freezing or storage at −80 °C, are commonly used to prevent nucleic acid degradation and maintain their integrity, which will preserve the genetic material before downstream analysis [55].

The pH level of the sample can substantially impact the effectiveness of enzymatic methods. DNase and RNase enzymes optimally function at neutral pH values of 5.0 and 7.6, respectively [56]. If pH values are too high or low, enzymatic activity may be substantially reduced or denatured, ultimately affecting nucleic acid removal performance. Therefore, adjusting the pH of the sample within the optimal range for the specific enzyme is essential for ensuring effective enzymatic pretreatment. Highly acidic or alkaline conditions can lead to nucleic acid degradation over time. To preserve genetic material in wastewater samples, maintaining a neutral or slightly acidic pH is often preferred.

The concentration of dissolved salts, or salinity, in wastewater samples can affect pretreatment methods for molecular detection. High salinity levels can inhibit enzymatic activity, potentially reducing nucleic acid removal efficiency [57]. To optimize the enzymatic pretreatment, adjustments to salinity levels may be necessary, including sample dilution or the addition of buffers to control salinity. Salinity can also influence nucleic acid extraction and purification methods by affecting binding and elution. Adjustments to extraction buffers or methods may be required to guarantee effective nucleic acid recovery.

The presence of organic matter, such as proteins, lipids, and humic acids, can significantly impact pretreatment methods. Similar to salinity, organic matter can interfere with enzyme activity and reduce nucleic acid removal performance. Additional enzyme treatments or enzymatic inhibitors may be needed to mitigate this interference. Organic matter can co-precipitate with nucleic acids during extraction and purification, affecting yield and purity [58]. Special extraction methods or reagents may be required to alleviate these effects. Moreover, suspended solids and particulates that contain organic matter can clog filtration membranes and interfere with centrifugation. Prefiltration steps can help reduce organic matter interference in downstream pretreatment.

## 5. Challenges and Limitations

### 5.1. Variability in Wastewater Composition

The composition of wastewater can vary significantly depending on its source and environmental conditions, which will impact the effectiveness of pretreatment methods. Wastewater composition can diverge widely between different sources and sampling points. As a result, it becomes challenging to establish a uniform, one-size-fits-all pretreatment method. Variability in organic matter content, pH, and salinity directly influences enzyme activity, which affects the performance of enzymatic pretreatment methods. Additional optimization may be necessary for each sample to guarantee the efficiency and consistency of enzymatic pretreatment across different wastewater samples [57–59].

Wastewater may contain various contaminants, including heavy metals, chemicals, and inhibitors, which have the potential to interfere with pretreatment methods and molecular assays. Contaminants in wastewater can limit the effectiveness of pretreatment methods, leading to false molecular detection results. Certain pretreatment methods exhibit selectivity for specific contaminants, nucleic acids, or microorganisms. However, variations in wastewater composition can compromise this selectivity, potentially allowing interference from non-target substances [60]. The selectivity of pretreatment methods may be lower when confronted with the unpredictability and diversity of wastewater samples, ultimately affecting the precision and specificity of molecular detection.

### 5.2. Potential Risk of Cross-Contamination

Cross-contamination represents a significant challenge and limitation in the use of pretreatment methods. It can occur during various stages, such as sample handling, transportation, and transformation from one sample to another, ultimately affecting the precision and reliability of pathogenic detection [61]. Notably, cross-contamination during extraction can lead to the inadvertent co-extraction of nucleic acids from different samples, resulting in false results, particularly for low-level pathogen samples. Thus, implementing strategies to mitigate cross-contamination, such as decontamination procedures or routine cleaning, and using dedicated equipment for pretreatment, becomes imperative.

Proper handling and storage practices for wastewater samples are essential to prevent cross-contamination. Improper handling procedures may contaminate equipment, work surfaces, or adjacent samples. Maintaining a controlled environment for sample handling and storage often requires additional resources and personnel training to minimize the risk of cross-contamination. The appearance of positive and negative controls in experiments is vital to validate the absence of cross-contamination and the accuracy of molecular detection results [62]. While the use of controls increases the complexity of experiments and the requirement for additional samples and resources, it is necessary to include them to ensure data integrity.

### 5.3. Potential Loss of Target Nucleic Acids

The potential loss of target nucleic acids in pretreatment methods can have several significant effects on the overall molecular detection process. Mainly, extraction methods may not effectively capture all target nucleic acids from the sample due to low nucleic acid yield or loss during extraction steps, resulting in the loss of target genetic material. If a significant portion of the target genetic material is lost, the assay may fail to detect pathogens at low concentrations in the wastewater sample [27]. Consequently, false negatives may occur where pathogens are present undetected, resulting in an underestimation of contamination levels. The potential loss of target nucleic acids can also lead to the inaccurate quantification of target pathogens, as the assay may not faithfully represent the actual pathogen load in the wastewater. This misinterpretation can lead to incorrect risk assessments regarding the presence of pathogens in the wastewater.

Furthermore, nucleic acid loss can introduce non-specificity into the detection process. Contaminants or interfering substances that co-purify with target nucleic acids may influence assay results. This influence can lead to false positives due to the presence of non-specific amplification products or signals, potentially resulting in the misidentification of pathogens or overestimation of contamination levels. Therefore, additional resources and higher sample volumes may be required to ensure adequate pathogen detection, which can increase operational costs and extend the time needed for analysis.

### 5.4. Environmental and Cost Considerations

Several pretreatment methods, especially chemical treatments, can exert a significant environmental impact due to the potential generation of chemical reagents and waste. Energy-intensive pretreatment approaches can contribute to a higher environmental impact, specifically if the energy source is derived from non-renewable fossil fuels [63]. Increased energy consumption, particularly from fossil fuels, can lead to higher greenhouse gas emissions, thereby negatively influencing the environment and contributing to climate change. Employing pretreatment methods with a substantial environmental footprint may conflict with sustainability goals and regulatory requirements for reducing environmental harm.

Moreover, the safe handling and disposal of chemicals can increase the operational costs and environmental impact associated with pretreatment methods. Some techniques, such as ultrasonication, can be energy-intensive, thereby escalating operational costs and environmental implications. The implementation of molecular methods for microbiological assessment in wastewater may lead to extra costs, including initial expenses for instruments, the regular maintenance and calibration of equipment, training, and data

analysis software. The high energy cost may present limitations, especially for wastewater treatment facilities or research projects with limited budgets. Therefore, a comprehensive consideration of the environmental consequences and costs associated with pretreatment methods is essential for making informed choices that align with sustainability objectives and financial constraints.

## 6. Conclusions and Future Research

In conclusion, this comprehensive review has explored the diverse prospects of pretreatment methods aimed at removing free nucleic acids from wastewater samples, ultimately ensuring the accuracy and reliability of pathogenic detection. Filtration, centrifugation, ultrasonication, chemical treatments, and enzymatic processes have demonstrated their effectiveness in removing free nucleic acids. However, their performance is significantly influenced by factors such as wastewater type and composition, pathogen type, and environmental conditions.

Despite the remarkable progress made by various pretreatment methods in addressing the issue of nucleic acid removal, several challenges still exist. Notably, the potential for nucleic acid loss, especially in some aggressive chemical and physical treatments, highlights the need for precise optimization. Additionally, environmental concerns, including eco-toxicity and resource depletion, are driving a shift toward more sustainable and eco-friendly practices. Cost considerations, including operational, resource, waste management, or microbiological assessment of costs, remain a crucial factor in determining the practicality and scalability of these approaches. This review underscores the importance of selecting pretreatment methods with careful attention to their environmental sustainability and cost-effectiveness, as these factors are central to the long-term viability of wastewater pathogenic detection.

The future directions for pretreatment methods in removing free nucleic acids in wastewater samples hold substantial promise as the field continues to evolve and address various challenges. Emerging technologies, such as advanced filtration membranes, nanomaterial-based approaches, and innovative enzyme methods, show potential in overcoming the current limitations. The combination of dye and enzymatic pretreatment is a promising alternative owing to its potential for synergistic and highly effective treatment. However, further research is needed to investigate the mechanisms and validate their viability. The direct integration of pretreatment methods into the sample collection process is an area that requires more exploration. This approach has the potential to reduce the risk of DNA degradation during transportation and storage and provide a more streamlined workflow for pathogenic detection in the field.

Moreover, interdisciplinary collaboration between microbiologists, environmental scientists, engineers, and public health experts is essential to drive innovation and address the complexities of wastewater pathogenic detection. Such collaborative efforts can lead to innovative and holistic solutions for the removal of free nucleic acids in wastewater while preserving the integrity of target DNA. By navigating these challenges and embracing emerging technologies, the field of pretreatment methods for wastewater pathogenic detection is poised to make significant contributions to public health and environmental protection in the future.

**Author Contributions:** Conceptualization, K.A.V.; writing—original draft preparation, K.A.V.; writing—review and editing, K.A.V., T.A.N. and T.P.N. All authors have read and agreed to the published version of the manuscript.

**Funding:** This research received no external funding.

**Data Availability Statement:** Data are contained within the article, and are available on request from the corresponding author.

**Acknowledgments:** The authors would like to acknowledge the financial support of the ND-ECI Postdoctoral Fellowship Program.

**Conflicts of Interest:** The authors declare no conflict of interest.

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
