# Peer review of "Advances in Pretreatment Methods for Free Nucleic Acid Removal in Wastewater Samples: Enhancing Accuracy in Pathogenic Detection and Future Directions"

_2673-8007, doi:10.3390/applmicrobiol4010001_

Round 1
Reviewer 1 Report
Comments and Suggestions for Authors
The review article by Kien A. Vu and colleagues addresses a point of great importance to the field of water treatment and monitoring. The text is well written and organized in a way that makes it easy to understand, and provides good reference material for the evolution of studies in this area.
Reviewer 2 Report
Comments and Suggestions for Authors
A review article discussing the issue of detecting pathogens in wastewater. Currently, in times when water resources are very limited, the issue of water quality protection is very important. The presented work is a well-written review of methods for removing free nucleic acids, which are a factor that interferes with the detection of pathogens. The article is written based on the latest literature (only a few items are from before 2020).
I propose to make just a few minor corrections:
1. In the introduction, please write about traditional methods of detecting pathogens.
2. Have any comparative studies been carried out to assess the quality of sewage using traditional breeding methods on selective media and molecular methods? If so, please add one paragraph on this topic.
3. In the summary, please briefly mention the costs of microbiological assessment of sewage using molecular methods.
